# On Architecture Selection for Linear Inverse Problems with Untrained Neural Networks

**DOI:** 10.3390/e23111481

**Published:** 2021-11-09

**Authors:** Yang Sun, Hangdong Zhao, Jonathan Scarlett

**Affiliations:** 1Department of Computer Science, National University of Singapore, 15 Computing Dr., Singapore 117418, Singapore; dcszhdg@nus.edu.sg (H.Z.); scarlett@comp.nus.edu.sg (J.S.); 2Department of Mathematics, National University of Singapore, 10 Lower Kent Ridge Rd., Singapore 119076, Singapore; 3Institute for Data Science, National University of Singapore, 3 Research Link, Singapore 117602, Singapore

**Keywords:** linear inverse problems, untrained neural networks, compressive sensing, deep decoder, architecture design, hyperparameters

## Abstract

In recent years, neural network based image priors have been shown to be highly effective for linear inverse problems, often significantly outperforming conventional methods that are based on sparsity and related notions. While pre-trained generative models are perhaps the most common, it has additionally been shown that even untrained neural networks can serve as excellent priors in various imaging applications. In this paper, we seek to broaden the applicability and understanding of untrained neural network priors by investigating the interaction between architecture selection, measurement models (e.g., inpainting vs. denoising vs. compressive sensing), and signal types (e.g., smooth vs. erratic). We motivate the problem via statistical learning theory, and provide two practical algorithms for tuning architectural hyperparameters. Using experimental evaluations, we demonstrate that the optimal hyperparameters may vary significantly between tasks and can exhibit large performance gaps when tuned for the wrong task. In addition, we investigate which hyperparameters tend to be more important, and which are robust to deviations from the optimum.

## 1. Introduction

Linear inverse problems arise in a wide range of application domains, such as computational imaging, optics, and remote sensing. Broadly, the problem consists of measuring a target signal $x^{*}\in R^{n}$ via linear measurements of the form
(1)$$y=Ax^{*}+w,$$
where $y\in R^{\ell }$ is the dimensionality-reduced observation vector, $A\in R^{\ell \times n}$ is a linear operator that captures the forward process, and $w\in R^{\ell }$ represents additive noise. The aim is to recover (an estimate of) the unknown signal $x^{*}$ given y and A. This setup captures a variety of problems including inpainting, denoising, super-resolution, and compressive sensing.

To permit the accurate recovery of $x^{*}$ with limited measurements (possibly ℓ≪n), a common approach in recent years has been to adopt explicit mathematical assumptions of low-dimensional structure, such as sparsity in a suitably-defined basis [1]. More recently, it has been observed that *data driven* generative priors can lead to considerable savings in the number of measurements [2], with a typical assumption being that $x^{*}$ can be well-approximated via a pre-trained generative neural network. Such pre-trained models tend to require large amounts of training data, which may be prohibitive in practical applications. In addition, even given large amounts of data, such solutions can suffer from distribution shift, since the training signals may not be fully representative of the test signals.

As an alternative method that can help overcome the above limitations, it has been observed that even *untrained* neural networks can serve as excellent priors for image recovery in linear inverse problems [3,4], where the input to the network is random and the weights are tuned to produce just a single image. In some cases, these techniques are combined with various forms of implicit or explicit regularization, such as early stopping [3], under-parametrized models [4], dropout methods [5], and total variation regularization [6]. Successful applications of the untrained approach have included recovery of natural images [3,4], magnetic resonance imaging [7], X-ray imaging [6], and computed tomography [8].

Broadly speaking, our work is motivated by the following gaps in the literature on inverse problems with untrained neural networks:Untrained neural networks invariably come with architectural hyperparameters (e.g., input size, number of layers, convolutional filters used, etc.), and while impressive results have been observed under various architectures, relatively little attention has been paid to how best to select these parameters.Untrained neural networks have primarily been applied in the context of recovering images (e.g., natural images, medical images, etc.), but to our knowledge, no detailed study has been given on how the architectural hyperparameters may vary across different signal types (e.g., rough vs. smooth), and different measurement types (e.g., inpainting vs. denoising vs. compressive sensing), nor on the robustness to changing from one setting to another.Regarding the signal type, while one-dimensional time-series data has been considered in numerous works on sparsity-based compressive sensing (e.g., neuro-electrical signals [9] and sensor network data [10]), to our knowledge, such signals have received significantly less attention in the context of neural network based priors, with one exception being a one-dimensional Deep Image Prior in [11].

Accordingly, in this paper, focusing on the Deep Decoder approach [4,12], we seek to further explore the utility of untrained neural networks beyond image recovery (in particular, to time-series signals), and more importantly, to better understand the role of architecture selection when recovering signals from different measurement models and signal types. We present algorithms for automatically tuning the architectural hyperparameters, and experimentally observe their behavior in diverse settings of interest. We focus in particular on addressing the following questions:(i)To what extent do the optimal hyperparameters vary for different measurement models and signal types?(ii)To what extent does the performance degrade when the hyperparameters are tuned for one setting but applied to another?(iii)To what extent are the various hyperparameters robust to deviations from their optimal value?

Regarding question (i), we find that the optimal configurations can depend heavily on both the measurement models and the signal types. Accordingly, it is natural in question (ii) that transferring settings can degrade the performance significantly, and we find that this is indeed commonly observed (though not always the case). Finally, regarding question (iii), we identify both examples of robust and non-robust behavior under deviations from the optimal value. These findings are based on experimentation on a diverse range of synthetic and real-world data types, with a particular emphasis on one-dimensional time-series signals.

### 1.1. Related Work

The literature on neural network techniques in inverse problems is rapidly growing [2,3,4,13,14,15,16]. We refer the reader to [17] for a recent survey, and focus here on the most closely-related works.

Two prevailing approaches in the literature are Deep Image Prior [3] and Deep Decoder [4], which adopt a similar high-level approach of tuning network weights to produce a single image. As outlined above, in contrast with approaches based on pre-trained models learned from data [2], Deep Image Prior and Deep Decoder are untrained, and consist of fitting neural network weights to a single image. Follow-up works have also studied different interpretations of Deep Image Prior from the perspective of architecture regularization [18] and Maximum a Posteriori Probability (MAP) estimation [19]. Although concerns have been raised that these approaches may lose information during their intermediate layers [20]; both have shown excellent (and typically comparable) performance in various inverse problems. In many cases, Deep Decoder enjoys the additional benefit of not requiring early stopping due to its relatively simple structure. In this paper, we focus our attention on Deep Decoder, but an analogous investigation of Deep Image Prior may be of interest in future work.

After the introduction of the Deep Decoder model in [4], several follow-up works explored variations and applications, which we outline as follows:In [12], several variants of Deep Decoder are introduced depending the presence/absence of upsampling and certain convolution operations. The success of Deep Decoder is primarily attributed to the presence of convolutions with fixed interpolating filters in the neural network architecture. Further details are given in Section 1.2.Theoretical guarantees for compressive sensing were given in [14,21]. The former studies the convergence of a projected gradient descent algorithm in underparametrized settings, whereas the latter shows that regular gradient descent is able to recover sufficiently smooth signals even in overparametrized settings.Variations of Deep Decoder for medical imaging are given in [7,22], with an additional challenge being combining measurements from multiple coils measuring the same signal in parallel. Additional applications of deep decoder include quantitative phase microscopy [23] and image fusion [20]. In addition, another variant of Deep Decoder for graph signals is given in [24].In [25], a method is proposed for combining the benefits of trained and untrained methods, by imposing priors that are a combination of the two.In [26], various robustness considerations for neural network based methods (both trained and untrained) are investigated. In particular, (i) both are shown to be sensitive to adversarial perturbations in the measurements; (ii) both may suffer from significant performance degradation under distribution shifts; and (iii) evidence is provided that the overall reconstruction performance is strongly correlated with the ability to recover specific fine details.

The importance of hyperparmeter selection was highlighted in [7], but has remained relatively unexplored. The above mentioned more recent work [26] developed independently from ours, and at least one of their observations therein matches one of ours (namely, performance degradation when the hyperparameters are tuned for the wrong data type), but, overall, our work and [7,26] remain mostly separate. Regarding the signal type, in our understanding, the recovery of one-dimensional signals has received much less attention compared to two-dimensional images, with one exception being the application of Deep Image Prior to time series signals in [11].

Architecture search is a popular topic in machine learning, particularly in the context of classification problems. Proposed search methods include reinforcement learning [27], parameter sharing [28], and differentiable search [29]; see [30] for a survey. However, to the best of our knowledge, such techniques cannot easily be applied in the context of signal recovery with untrained neural networks, which is notably distinct from the supervised problem of classification. Instead, our approach will use the simpler idea of searching over *parametrized* architectures, building on hyperparmeter optimization techniques [31,32]. Having said this, we believe that adapting non-parametrized architecture search methods to our setting could be of significant interest in future work.

### 1.2. Background: Deep Decoder

The Deep Decoder, denoted as G(C) and parameterized by model weights C, transforms a randomly chosen and fixed input tensor $B_{1}\in R^{n_{1}\times k_{1}}$, which is the “latent” input consisting of $k_{1}$ many $n_{1}$-dimensional channels, to an $n_{d}\times k_{out}$ dimensional image x. Here, *d* is the depth (i.e., number of layers) of the network. The network transforms the tensor $B_{1}$ to an image using batch normalization [33] (which is equivalent to channel-wise normalization here), upsampling operations, pixel-wise linearly combining of channels, and rectified linear units (ReLUs). In the original version of the Deep Decoder [4], the tensors in the (i + 1)-th and final layer are given by
(2)$$\begin{array}{cc} \;B_{i+1}= & \mathrm{bn}(\mathrm{relu}\left(U_{i}B_{i}C_{i}\right)),\;\;i=1,\ldots ,d-1,\;\\ \;x= & \mathrm{sigmoid}(B_{d}C_{d}),\; \end{array}$$
where the coefficient matrices $C_{i}\in R^{k_{i}\times k_{i+1}}$ are the network weights of 1×1 convolutions kernels on layer *i*, bn() is batch-normalization, and $U_{i}\in R^{n_{i+1}\times n_{i}}$ is an upsampling tensor. The model architecture is depicted in Figure 1 in the context of producing 2D images; for 1D signals, the structure is similar with two dimensions “flattened” into one.

Several variants of Deep Decoder were presented in [12], depending on the presence of upsampling and fixed vs. trained convolutional kernels. These models are all closely related, and we focus on a particular one (termed “model (i)” in [12]), which we found to consistently perform best (or equal best) in terms of computation time and recovery performance.

In the model that we focus on, for each layer, an additional fixed-kernel convolution operation is performed before the 1×1 convolution described above. Specifically, we have $U_{d}=I$, all other $U_{i}$ still as upsampling operators, $C_{i}\in R^{k_{i}\times k_{i+1}}$ same as the above, but plus an additional operator $T(c_{i})$ performing a convolution with the fixed-kernel $c_{i}\in R^{s}$ (where *s* is the size of kernel and is a hyperparameter which we will explore in this paper) on layer *i*. For instance, in the 1D setting, the following filter of length s=4 could be used:(3)$$c_{i}=\frac{1}{16}\left[\begin{array}{cccc} 3 & 9 & 9 & 3 \end{array}\right].$$
A filter size of 4 is indeed suggested in [12], but we also allow for other sizes. For a general size *s*, we define the center to be $\frac{s-1}{2}$, and let the *i*-th filter weight be proportional to $\exp(-2\;{\mathrm{dist}}_{i})$, where ${\mathrm{dist}}_{i}$ is the distance to the center. The filter weights are always normalized to sum to one.

Given the Deep Decoder network G(C), the problem of estimating $x^{*}$ given (A,y) is performed as follows: (4)$$\hat{C}\leftarrow \underset{C}{\mathrm{minimize}}\;{∥AG(C)-y∥}^{2},\;\hat{x}=G\left(\hat{C}\right),$$
where the “minimize” operation may not necessarily correspond to finding a global minimizer, but rather corresponds to running a variant of Gradient Descent for a certain number of iterations.

Our general goal is for $\hat{x}=G\left(\hat{C}\right)$ to be a good approximation of $x^{*}$. In general, we may measure the performance according to some generic loss function $\ell (x,\hat{x})$. For concreteness, we will focus primarily on the squared loss, $\ell \left(x,\hat{x}\right)={∥x-\hat{x}∥}^{2}$, and, in our experiments, we will reparametrize this according to the more widely-adopted Peak Signal-to-Noise Ratio (PSNR):(5)$$\mathrm{PSNR}=20\log_{10}\left(\frac{{\mathrm{MAX}}_{I}}{\sqrt{\frac{1}{n}{∥x-\hat{x}∥}^{2}}}\right),$$
where ${\mathrm{MAX}}_{I}$ is the maximum possible signal value. The PSNR serves as a natural measure for reconstruction accuracy is ubiquitous in the signal processing literature, and has the desirable feature of being invariant to rescaling (e.g., when converting from [0, 1]-valued images to [0, 256]-valued images). On the other hand, our proposed techniques are general and could be used alongside other measures such as the structural similarity index (SSIM).

### 1.3. Hyperparameters and Problem Variables

In view of the above description of Deep Decoder, we focus on optimizing the following important hyperparameters: (i) input tensor size, (ii) number of channels per layer, (iii) number of layers, (iv) filter size for the fixed-kernel, and (v) step size in the Adam optimizer. While further hyperparameters could also be considered (e.g., activation function, other optimization parameters), we found these to consistently work well with fixed choices.

We note that the preceding hyperparameters also implicitly determine the upsampling factor. Specifically, with the output size being fixed according to the problem, further fixing the input size and number of layers also fixes the upsampling factor (assumed to be the same in each layer, up to rounding). In contrast, previous works focused on an upsampling factor of two.

The problem variables that can potentially impact the choice of hyperparameters include the signal type (e.g., slow vs. fast varying), signal length, measurement type (e.g., inpainting, compressive sensing, etc.), compression ratio $\frac{\ell }{n}$, noise level, and so on.

## 2. Hyperparameter Selection

In this section, we present two simple and general-purpose algorithms for optimizing architectural parameters, as well as giving some theoretical insight based on statistical learning theory.

We consider a setup in which, for a particular measurement matrix A, we have access to a data set $D={\left\{\left(x_{j},y_{j}\right)\right\}}_{j=1}^{m}$, (Our analysis and algorithms also equally apply when A is random and the data set takes the form $D={\left\{\left(x_{j},A_{j},y_{j}\right)\right\}}_{j=1}^{m}$.) where each $y_{j}$ is produced from the corresponding $x_{j}$ according to (Equation 1). We consider the case that the architecture is parametrized by a list *H* of hyperparameters (e.g., input size, number of layers, etc.), and we write $G\left(C\right)=G_{H}\left(C\right)$ to highlight this dependence. For any specific choice of *H*, we can form $\hat{x}$ according to (Equation 4), and measure the performance according to the loss function $\ell (x,\hat{x})$. Our goal is to use the training data D to find a good choice of *H*. To provide insight on this task, we first consider a theoretical viewpoint.

### 2.1. Theoretical Viewpoint

Consider a statistical learning setup, in which the training examples in D are assumed to be independently drawn from some unknown distribution $P_{XY}$ (where $P_{Y|X}$ follows (Equation 1) with a fixed measurement matrix A, but the noise distribution is unknown). For convenience, we consider the loss as a function of (x,y):(6)$$\gamma_{H}\left(x,y\right)=\ell \left(x,\hat{x}\left(H,y\right)\right),$$
where $\hat{x}\left(H,y\right)$ is the estimated obtained by running (Equation 4) with hyperparmeters *H* (and with the known measurement matrix A). In this setting, there is a precise notion of the “best” hyperparameter configuration:(7)$$H^{*}=\underset{H\in H}{\arg\min}E\left[\gamma_{H}\left(x,y\right)\right],$$
where (x,y) is a fresh sample from $P_{XY}$. While $H^{*}$ cannot be computed directly (due to $P_{XY}$ being unknown), we can adopt the widespread idea from statistical learning theory of minimizing the *empirical loss*:(8)$$\begin{array}{cc} \;\hat{H} & =\hat{H}\left(D\right)=\underset{H\in H}{\arg\min}\frac{1}{m}\sum_{j=1}^{m}\gamma_{H}\left(x_{j},y_{j}\right).\; \end{array}$$
As well as providing the starting point for practical techniques in subsequent subsections, this empirical loss minimization approach is theoretically principled, giving the following theoretical guarantee analogous to standard PAC-learnability results (e.g., ([34] Section 4.2)).

**Theorem** **1.**
*If H takes values within a finite set H, and the losses are bounded by (This extends to arbitrary bounds of the form $\gamma_{H}\left(x,y\right)\in \left[a,b\right]$ by rescaling.) $\gamma_{H}\left(x,y\right)\in \left[0,1\right]$ for all (x,y), then for a training set D of size m and any η>0, we have with probability at least 1−η that*

(9)
$$E\left[\gamma_{\hat{H}}\left(x,y\right)\right]\leq E\left[\gamma_{H^{*}}\left(x,y\right)\right]+\sqrt{\frac{2}{m}\left(\log|H|+\log\frac{2}{\eta }\right)}.$$



The proof follows standard statistical learning theory arguments and can be found in the Appendix A. This result indicates that, to approximate $H^{*}$, it suffices to have a number of training signals growing as m=O(logH). In particular, if there are *K* hyperparameters taking a bounded number of values each, this reduces to m=O(K), a linear dependence on the number of hyperparameters. However, it is important to note that this is a *worst-case* guarantee, and in Section 3 we will demonstrate the effectiveness of approximately solving (Equation 8) even with a very small number of training signals.

In the remainder of the section, we explore practical algorithms for approximately solving (Equation 8). We note that, even if each hyperparameter is restricted in advance to take finitely many values, a brute-force evaluation of all configurations is typically prohibitive, since even a single evaluation requires a separate optimization to solve (Equation 4). Hence, the goal is to efficiently explore a *subset* of configurations to find an *approximate* solution to (Equation 8).

### 2.2. Successive Halving

To approximately solve (Equation 8), we first utilize the idea of Successive Halving [31]. The details are given in Algorithm 1, in which the algorithm takes as input the number of configurations *n*, a minimum resource level *r* and maximum resource level *R* (e.g., number of optimization iterations; see below for details), and a reduction factor η≥2. The algorithm makes use of the following simple subroutines:The function random_configurations(n) generates a set of *n* random configurations, i.e., for each such configuration, each hyperparameter value is chosen uniformly at random from the pre-specified finite set of possible values.The function $\mathrm{evaluate}\_\mathrm{psnr}(H,r_{i})$ returns the PSNR after running the minimization (Equation 4) with the hyperparameter configuration *H* and resource level $r_{i}$.The function $\mathrm{top}\_k(H,P,n_{i}/\eta )$ finds the $n_{i}/\eta$ highest values of PSNR in the list P, and returns the corresponding $n_{i}/\eta$ configurations in H.

Successive Halving uniformly allocates a budget to a set of hyperparameter configurations, evaluates the PSNRs of all configurations, keeps the top 1/η, and increases the budget per configuration by a factor of η. This repeats until the maximum per-configuration budget of *R* is reached, and only one configuration remains. The algorithm allocates exponentially more resources to more promising configurations. The resource under consideration could be the number of iterations of stochastic gradient descent, the number of training examples, and the number of random features, etc.; in this paper, we focus on the number of iterations of gradient descent.
**Algorithm 1:** Successive Halving for Hyperparameter Optimization.
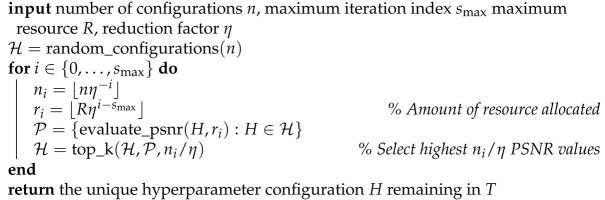


### 2.3. Greedy Fine-Tuning

Even after running Successive Halving, it may still be beneficial to perform some fine-tuning operations to obtain a potentially improved hyperparameter configuration. Here, we introduce another simple greedy algorithm to perform such fine-tuning, detailed in Algorithm 2. The algorithm takes as input the baseline configuration *H*, and iteratively updates one hyperparameter at a time by choosing whether to slightly increase, slightly decrease, or remain the same. (We do not have any categorical variables in our experiments, but if any were present, they could be handled similarly, e.g., by trying every category or a random subset.) This is continued until a stopping criteria is met (e.g., maximum number of iterations, or every parameter stayed the same).

Let the number of hyperparameters be denoted by *K*. The fine-tune levels $P_{1},\ldots ,P_{K}$ are user-specified, and may be iteration-dependent; the approach we use will be described in Section 3.2, along with our stopping condition. Once these are selected, the procedure simply iterates through the *K* hyperparameters; for each one, we evaluate the PSNR of the two new candidate values (with all other hyperparameters held fixed) in the $\mathrm{finetune}\_\mathrm{single}(H,P_{i})$ subroutine and identify the highest PSNR among the three options (the third option being to remain the same), and update accordingly.
**Algorithm 2:** Greedy Fine-Tuning for Hyperparameter Optimization.
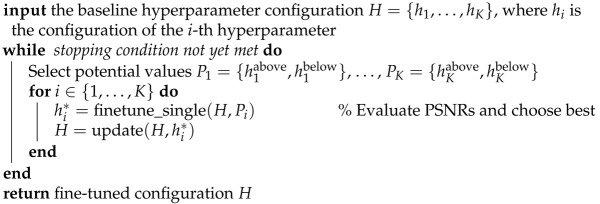


## 3. Experiments

In this section, we present various experimental findings based on the algorithms proposed on Section 2. We consider a variety of measurement models and signal types described in Section 3.1, as well as further considering natural signals and images in Section 3.6, where we compare with existing baselines. Overall, our experiments are chosen to cover a diverse range of settings representative of those considered in previous works such as [2,4,11], but we emphasize that our focus is on general-purpose methods, and accordingly, we do not seek to compete against highly specialized techniques on a case-by-case basis. Our code is available at https://github.com/ethangela/compressd-sensing (6 October 2021).

### 3.1. Measurement Models and Signals

We consider a variety of measurement models of the general form (Equation 1), focusing on the noiseless scenario except where stated otherwise:For *random inpainting*, a given fraction of the signal values are masked (i.e., not observed), and we consider the fractions 9/10, 3/4, 1/2, 1/4, and 1/8.For *block inpainting*, the signal is divided into blocks (taken to be of length 16 for all 1D signals and of block size 8 × 8 for all 2D signals), and a given fraction of the blocks are masked (i.e., not observed).For *compressive sensing*, we consider A taking the form of randomly subsampled Gaussian circulant measurements, e.g., see [35]. Such matrices can be viewed as approximating the behavior of i.i.d. Gaussian matrices (e.g., as considered in [2]), but with considerably faster matrix operations. We mostly use 100, 500, or 1024 measurements, but sometimes also consider other values.For *denoising*, A is the identity matrix (i.e., every entry is observed), but the noise term in (Equation 1) corrupts the measurements. We consider i.i.d. Gaussian noise with standard deviations 0.05 and 0.1.

To permit the consideration of distinct signals with a clear notion of lying in a given “class”, we consider synthetic signals drawn from a Gaussian Process (GP) [36] of length n=4096. We consider a class of “rough” signals and a class of “smooth” signals on the domain [0,1], with the former being drawn from a GP with an exponential kernel of lengthscale $\frac{1}{80}$, and the latter being drawn from a GP with radial basis function (RBF) kernel of lengthscale $\frac{1}{20}$. Example signals are shown in Figure 2.

Whenever we report PSNR values, these are evaluated on *test signals* that are generated separately from the training signals. Specifically, we average over five such test signals to obtain the average PSNR. We also re-generate the random A for these test signals, to ensure that we are measuring the generalization to different measurement matrices from the same family.

### 3.2. Algorithmic Details

We apply Successive Halving (Algorithm 1) to optimize the following hyperparameters, initially limiting to the following values:#layers ∈[2,4,6,8,10,14,18,22,26,30].#channels ∈[16,32,64,128,192,256,320,384,448,512].input_size ∈[2,4,8,16,24,32,48,128,256,512].filter_size ∈[4,8,16,32,48,64].step_size ∈[0.0005,0.001,0.002,0.003,0.005,0.007,0.01,0.025,0.05].

We let the resource in Algorithm 1 be the number of optimization iterations with a maximum value of R=2048. The number of possible configurations produced from the five hyperparameter sets above is the cardinality of their Cartesian product, 54,000, and we reduce this to 5400 by taking a random 10% of them. We also set the reduction factor as η=3, and maximum iteration index as $s_{\max}=8$. The Adam optimizer parameters other than step_size are kept at their default values.

For greedy fine-tuning (Algorithm 2), we choose set $p_{\mathrm{above}}$ and $p_{\mathrm{below}}$ to be of the form $p_{\mathrm{current}}\pm \Delta$, where Δ is initially chosen to be half the distance to the nearest value in the corresponding list above, and is subsequently halved on each iteration. We stop updating when the distances fall down to 1, except for the non-integer parameter step_size, for which we use a threshold of 0.0005.

### 3.3. Optimized Parameters for Varying Settings

In Table 1 and Table 2, we list the resulting optimized hyperparameters for GPs drawn from the exponential kernel and RBF kernel, respectively. These results provide evidence that the optimal hyperparameters can indeed vary significantly across measurement types and signal types. For instance, we see a pattern that more challenging tasks tend to favor more layers, with the exponential kernel signals (less smooth) tending to choose many more than the RBF kernel signals (more smooth), except random inpainting which is a relatively easier task. On the other hand, the input size tends to be smaller for harder tasks (e.g., block 9/10) and higher for easier tasks (e.g., random 1/8). Our results also suggest that a moderate number of channels (e.g., 150 to 200) and a relatively small step size (e.g., 0.003 to 0.005) tend to perform well across multiple tasks.

Naturally, it is important to not only consider the optimal values, but also the robustness with respect to varying values; this will be done in Section 3.5.

### 3.4. Cross-Performance and Transferability

To examine the importance of optimizing for the specific task at hand, we apply the optimized hyperparameter configuration of one type of measurement on other types of measurements, and compare the resulting PSNR values. The results are shown in Table 3 and Table 4 for the exponential and RBF signals, with columns indicating training measurement models and rows indicating test measurement models. The diagonals (in which the training and testing are matched to each other) are highlighted via shading.

The fact that the shaded (diagonal) entries have the highest (or very close to highest) value in each row verifies the importance of optimizing for the specific task at hand. In some cases, the gaps can be very significant, e.g., in the ‘compress 100’ row. While the shaded value can sometimes narrowly fall short of being best (e.g., ‘compress 500’ row), the gap is small, and is attributed to (i) the train and test signals still being different, despite being in the same class, and (ii) the possibility of two different tasks having similar optimal hyperparameters.

### 3.5. Effects of Single Hyperparameters

Next, we seek to examine the importance of single hyperparameters by taking the optimized configurations, and varying one hyperparameter at a time while keeping the rest fixed. While there are too many combinations of measurement models and hyperparameters to show here, representative examples are shown in Figure 3.

We find that a common feature in these plots is an “arch” shape, where the peak indicates the optimal value, though this is not always the case (e.g., for other signals/measurement models, filter_size was significantly more flat). A flat PSNR curve indicates robustness with respect to varying its value, whereas a highly varying PSNR curve indicates that the hyperparmeter is particularly important to optimize. For instance, the former scenario is observed in sub-figures (a,h).

We additionally comment that these curves can help identify the impact of over-estimating vs. under-estimating the optimal parameter choices for certain parameters. For example, sub-figures (d,g) indicate that decreasing the input size from its optimal value can degrade the performance much more compared to when increasing the value. Hence, if one were faced with uncertainty about exactly where the optimal value lies (e.g., due to possible future changes in the signal distribution or measurement type), one may prefer to favor a slightly higher input size in this example.

### 3.6. Real-World Data and Comparisons to Baselines

Here, we compare the optimized Deep Decoder to a variety of baselines, as well as moving from synthetic signals to real-world data.

For 1D real time-series data, we use hourly recordings of NO2 and O3 levels in air quality of an Italian town throughout 2004–2005 [37]. We split the recordings into signals of length n=1024 for O3 and n=512 for NO2, and normalize the range to [0,1]. As a data-cleaning pre-processing step, we trim the original signal to remove missing values, while “stitching together” the pieces in a manner that maintains continuity. Examples of the signals are shown in Figure 4.

For 2D image signals, we use the CelebA dataset [38] cropped to size 128×128×3. We note that, when considering CelebA data, the model structure is suitably adapted for 2D images. For example, the filter of fixed-kernel convolution introduced in Equation (Equation 3) is expanded from a vector to a square matrix, and similarly for the upsampling operations.

Similar to our synthetic experiments, we maintain a train/test separation for our real-world data experiments, optimizing the hyperparameters using the training signals but evaluating the performance on a *separate* set of test signals. For the 1D air quality data, the training size is 3 and the test size is 5, while for CelebA the training size is 5 and test size is 8.

We select four baseline models to compare against: Original Deep Decoder (Org. DD), original Deep Image Prior (Org. DIP), Lasso with the wavelet basis (LassoW), and Total Variation regularization (TV Norm). The first two methods are the pioneering untrained neural network priors discussed in Section 1 with fixed ‘default’ parameters, and the latter two methods are widely-used conventional priors. The hyperparameter configuration of Org. DD [12] is: #channels = 320 for inpainting and #channels = 128 for the other tasks, #layers = 6, input_size = 128, step_size = 0.01, and filter_size = 4. The hyperparameters of Org. DIP are sourced from the deep generative CNN architecture (U-Net) embedded in Org. DIP. We follow [3] and set the number of up/down layers as 5, the number of input channels as 32, and the number of channels for tensors in middle layers as 128. The number of optimization iterations for both Org. DD and Org. DIP is fixed to 10,000.

For LassoW and TV Norm, we optimize the hyperparameter for each type of signal and measurement, respectively. This hyperparameter, typically denoted λ, controls the weighting of the penalty to the loss function. The maximum number of optimization iterations for both LassoW and TV Norm is set as 2000.

Table 5 and Table 6 give the average PSNR for each estimator with selected measurements for GP signals, and Table 7, Table 8 and Table 9 give the results for NO2 data, O3 data, and CelebA images, respectively. We see that, throughout these experiments, Opt. DD is consistently either the best performing or very close to being so. The gains can be particularly significant for challenging tasks (e.g., compress 100 in Table 5 and compress 25 in Table 7 and Table 8), and for tasks with aspects most different from those that the existing methods were optimized for (e.g., DIP is less suited to RBF signals in Table 6).

The only case in which Opt. DD appears to underperform is denoising on 1D time-series data, in which the TV norm approach works unusually well (in comparison to its worse performance in many other cases). In one of these denoising entries, Opt. DD is in fact slightly below Org. DD, and we believe that this is due to the fact that having a small training set size can sometimes limit the generalization performance (e.g., if a test signal has some distinct features). However, among our diverse set of experiments, we otherwise found no such cases.

To complement the numerical values, we give some examples of reconstructed signal segments in Figure 5, Figure 6, Figure 7 and Figure 8. We observe that, at least in these examples, Opt. DD is able to better match the true signal with fewer artifacts. We also visualize the comparisons of reconstructed CelebA images among all baselines in Figure 9 and Figure 10, and we believe that they are consistently the most visually similar to the originals, thereby matching the consistent PSNR improvements observed in Table 9.

### 3.7. Accelerated Multi-Coil MRI Data

In this final experiment, we apply our proposed algorithms on data from Magnetic Resonance Imaging (MRI). MRI is a widely-used medical imaging technique that promises impressive accuracy, but is prone to requiring long scans that may cause discomfort. Starting with the pioneering work of [39], various compressive sensing based approaches have been proposed for taking fewer measurements and reducing the scan time.

In this experiment, we switch to a slightly more general observation model and consider a variant of the Deep Decoder specifically targeted at this application; we proceed by giving the relevant details. The main difference from the measurement model (Equation 1) is that measurements are taken from *multiple sensors* (coils). In addition, MRI machines are invariably constrained to take measurements in the Fourier domain. Thus, the goal is to recover an image $x^{*}\in C^{n}$ from a set of measurements of the form
(10)$$y_{i}=MFx^{*}+w,\;i=1,\ldots ,n_{c},$$
where $M\in R^{\ell \times n}$ is a mask that indicates which Fourier coefficients are observed, (Specifically, each row of M has a single 1 at the observed coefficient, and the remaining entries are zero.) $F\in R^{n\times n}$ is the Fourier transform operation, $n_{c}$ is the number of magnetic coils, and $w\in R^{\ell }$ still represents additive noise. Each $y_{i}\in R^{\ell }$ represents the *ℓ* measurements from a single coil, with the idea being that combining measurements from multiple coils improves accuracy. In general, each coil’s measurements may further be weighted according to a *sensitivity map* (i.e., some regions appear brighter than others), but this is omitted in (Equation 10) since we only seek to follow an experimental setup from [7] that similarly omitted sensitivity maps. The loss function in (Equation 4) is updated as follows:(11)$$\hat{C}\leftarrow \underset{C}{\mathrm{minimize}}\;\frac{1}{2}\sum_{i=1}^{n_{c}}{∥MFG\left(C\right)-y_{i}∥}^{2}.$$

We work with the recently-released fastMRI dataset [40] which consists of train/validation set of fully-sampled measurements of knees taken with $n_{c}=15$ coils. We replicate an experiment from [7] to test the performance of three different generators, Deep Image Prior, Deep Decoder, and Deep Decoder variant specifically called ConvDecoder which is specifically targeted towards the MRI application. Specifically, the difference is that, while Deep Decoder uses bi-linear upsampling and 1×1 convolutions, the ConvDecoder uses Nearest-Neighbor up-sampling and a 3×3 convolutional layer. We replace the constant 3 here by a tunable parameter filter_size.

In [7], Deep Decoder and ConvDecoder are tuned through a basic grid search, and our objective here is to tune these networks through our proposed algorithms and compare the results. The relevant additional details of this experiment are as follows:The mask M is chosen to be a standard 1D variable-density mask (i.e., random or equi-spaced vertical lines across the Fourier space), and is randomly chosen for each run. We do not add w explicitly, as the ground truth images already contain some noise.Following [7], three additional image comparison metrics are considered along with the PSNR, namely, the Visual Information Fidelity (VIF) [41], Structural Similarity Index (SSIM) [42], and Multi-Scale SSIM (MS-SSIM) [43]. However, following our previous sections, all hyperparameter tuning is done with respect to the PSNR.We optimize the same hyperparameters for ConvDecoder as our previous experiments, but the possible settings of filter_size for ConvDecoder during Successive Halving are adjusted as filter_size ∈ [2,3,4,5,6,7], in view of all considered filter sizes being small in [7]. For consistency with [7], we also change the number of optimization iterations to 20,000 for all methods.Following [7], we do not consider filter_size for Deep Decoder, and instead use the original Deep Decoder as introduced in (Equation 2), rather than the variant with an additional fixed-kernel convolution introduced just above (Equation 3).Each scan of a knee from fastMRI consists of a number of slices, each of which is a 2D image, and together the images form a 3D volume. We choose the the middle slice of the volume to obtain each image, and discard the other slices. The train and test sizes are set as 4 and 16, respectively.

Table 10 presents the evaluation performance of the optimized DD and ConvDD methods vs. the original variants and the original Deep Image Prior, where “original” means making use of the grid search hyperparameters from [7]. We observe that both Opt. ConvDD and Opt. DD are able to slightly improve on the original versions, again indicating the utility and versatility of our tuning algorithms. A minor exception is the SSIM, but the difference is very marginal; perhaps most important is the PSNR, since this is the metric that was optimized.

Two sets of sample reconstructions are shown in Figure 11. These reconstructions highlight that the visual quality can be improved in some cases (top row) but remains relatively unchanged in other cases (bottom row). The details of the hyperparamter configurations are also shown in the right part of Table 10. The most notable hyperparameter change from our tuning algorithm was the optimized step size, with a default value of 0.008 but our optimized value given by the smaller value of 0.002.

## 4. Conclusions

We have studied the role of architecture selection in signal recovery via untrained neural networks, with some of the main implications including (i) different measurement models and signals may benefit from significantly different hyperparameters; (ii) the performance may drop significantly when transferring configurations directly from one setting to another, and (iii) certain hyperparameters tend to exhibit better robustness to deviations from the optimum than others.

Possible directions for future work include performing a similar study of hyperparameter selection for Deep Image Prior [3], and perhaps more ambitiously, exploring methods that search *directly over architectures* rather than only hyperparameters of a pre-specified class of architectures (e.g., see [29] and the references therein in the context of neural networks for classification).

## Figures and Tables

**Figure 1 entropy-23-01481-f001:**
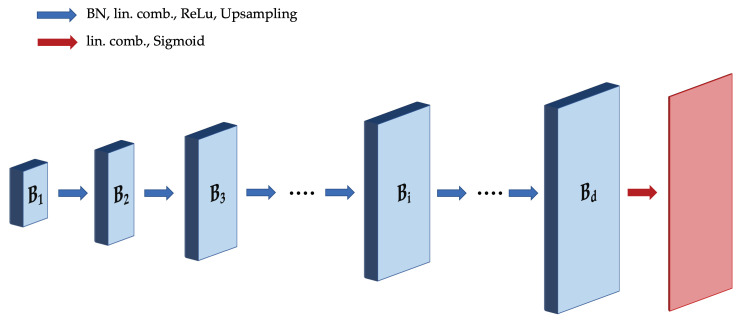
Illustration of Deep Decoder architecture for 2D signals. Each layer consists of channel-wise batch normalization, pixel-wise linear combining of channels (1×1 convolution), ReLU activation, and upsampling.

**Figure 2 entropy-23-01481-f002:**
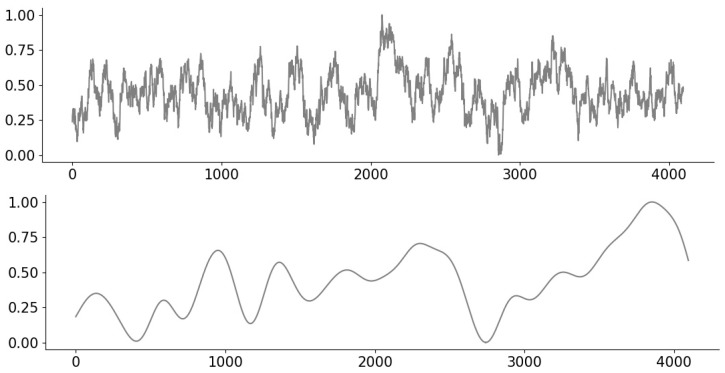
Example Gaussian Process (GP) signals corresponding to the exponential kernel (**top**) and RBF kernel (**bottom**).

**Figure 3 entropy-23-01481-f003:**
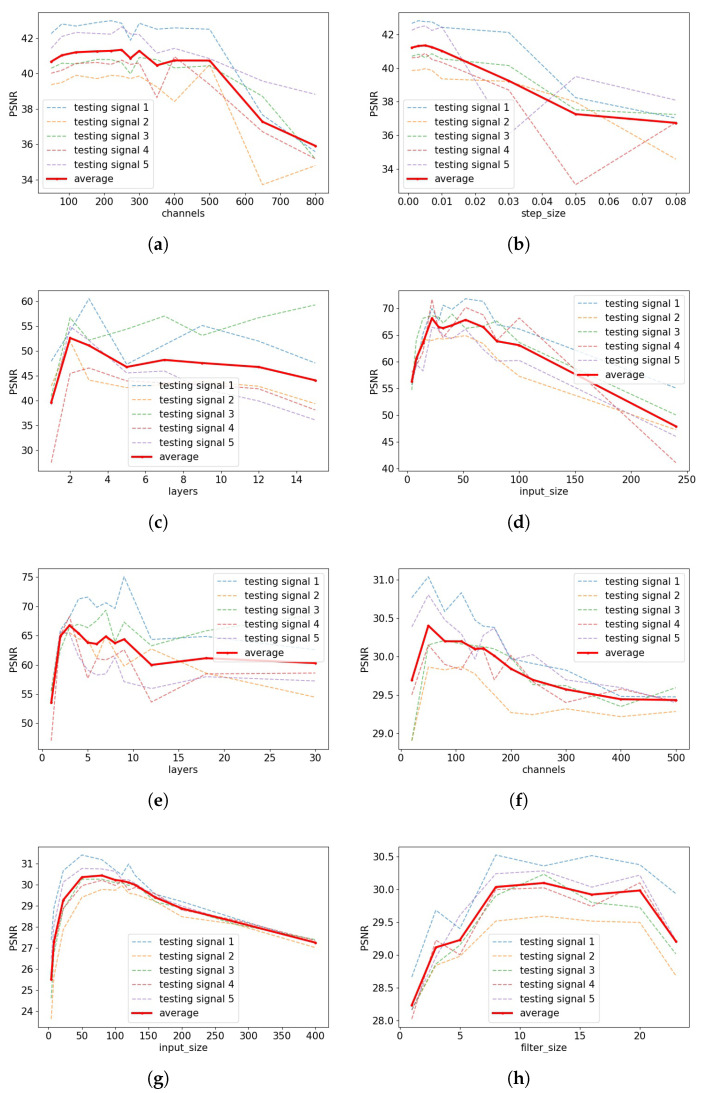
Example single-hyperparameter plots for various signals in the tasks of random inpainting (random 1/8), block inpainting (block 1/4), denoising (denoise 0.05), and compressive sensing (compress 500). (**a**) #channels, inpainting, exponential; (**b**) step_size, inpainting, exponential; (**c**) #layers, inpainting, RBF; (**d**) input_size, compressing, RBF; (**e**) #layers, compress, RBF; (**f**) #channels, denoising, exponential; (**g**) input_size, denoising, exponential; (**h**) filter_size, denoising, exponential.

**Figure 4 entropy-23-01481-f004:**
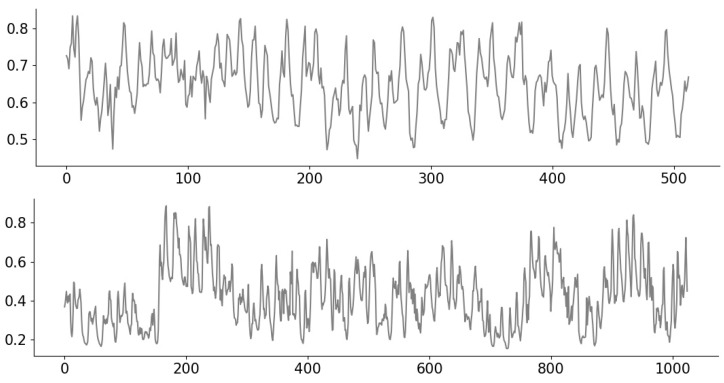
Examples of air quality time series signals for NO2 (**top**) and O3 (**bottom**).

**Figure 5 entropy-23-01481-f005:**
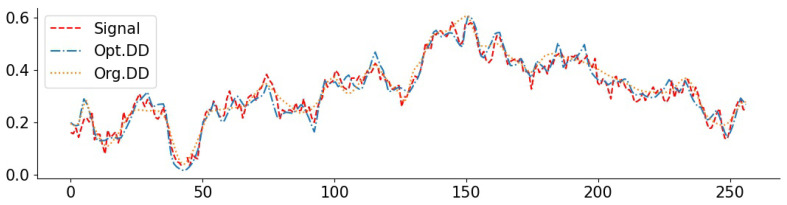
Exponential signal with denoising (0.05): PSNR(Opt. DD) = 29.79, PSNR(Org. DD) = 23.18.

**Figure 6 entropy-23-01481-f006:**
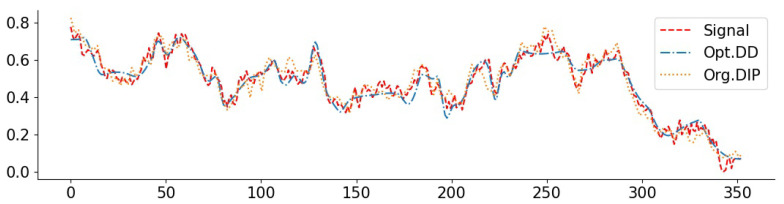
Exponential signal with compressing (1024): PSNR(Opt. DD) = 27.91, PSNR(Org. DIP) = 27.01.

**Figure 7 entropy-23-01481-f007:**
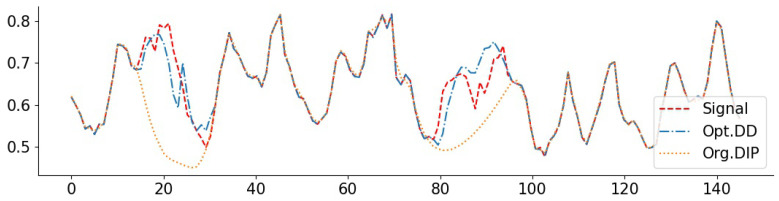
NO2 signal with inpainting (block 1/4): PSNR(Opt. DD) = 34.16, PSNR(Org. DIP) = 33.45. The large deviations of the Org. DIP curve correspond to locations of missing blocks.

**Figure 8 entropy-23-01481-f008:**
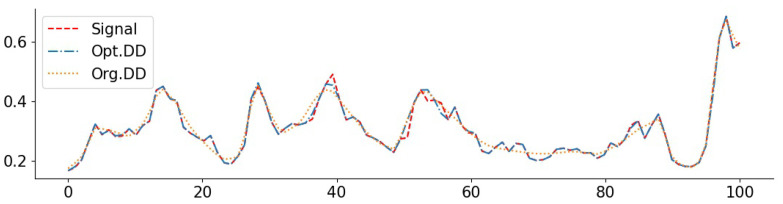
O3 signal with inpainting (random 1/8): PSNR(Opt. DD) = 37.26, PSNR(Org. DD) = 34.31.

**Figure 9 entropy-23-01481-f009:**
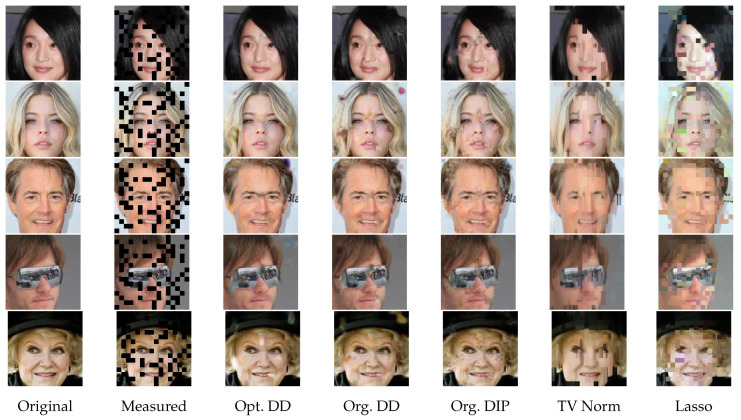
CelebA images with inpainting (block 1/4).

**Figure 10 entropy-23-01481-f010:**
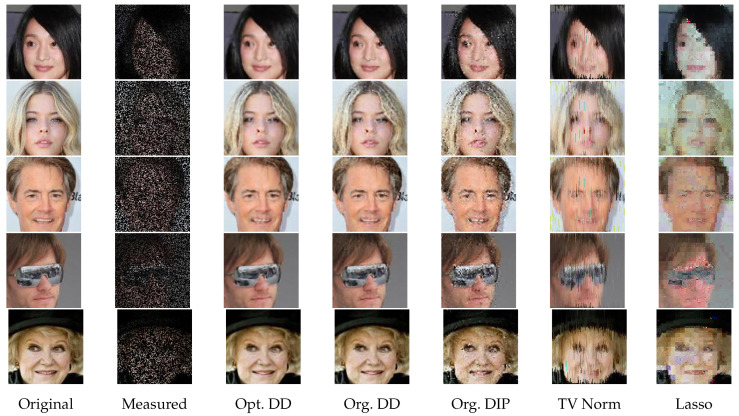
CelebA images with inpainting (random 3/4).

**Figure 11 entropy-23-01481-f011:**
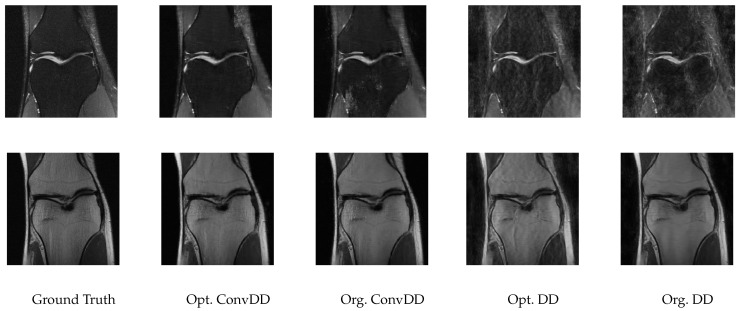
Sample reconstructions of multi-coil knee measurements from fastMRI images.

**Table 1 entropy-23-01481-t001:** Optimized hyperparameter configurations (exponential signals).

Measurement	#Channels	#Layers	Input_Size	Filter_Size	Step_Size
block 9/10	110	29	2	11	0.0030
block 3/4	158	25	5	12	0.0030
block 1/2	180	21	18	15	0.0030
block 1/4	180	15	40	30	0.0030
block 1/8	240	9	120	100	0.0070
random 9/10	196	11	45	73	0.0030
random 3/4	158	6	33	73	0.0030
random 1/2	200	4	512	73	0.0030
random 1/4	250	4	512	46	0.0030
random 1/8	250	8	512	2	0.0050
compress 100	83	24	21	11	0.0100
compress 500	146	20	22	9	0.0030
compress 1024	141	29	24	18	0.0030
denoise 0.1	128	10	63	29	0.0020
denoise 0.05	150	4	120	26	0.0070

**Table 2 entropy-23-01481-t002:** Optimized hyperparameter configurations (RBF signals).

Measurement	#Channels	#Layers	Input_Size	Filter_Size	Step_Size
block 9/10	301	3	9	27	0.0055
block 3/4	343	6	14	10	0.0025
block 1/2	227	6	12	37	0.0035
block 1/4	399	2	14	6	0.0100
block 1/8	244	2	38	20	0.0035
random 9/10	112	5	34	27	0.0040
random 3/4	147	7	34	16	0.0045
random 1/2	178	8	42	38	0.0030
random 1/4	166	8	108	26	0.0045
random 1/8	192	10	128	4	0.0025
compress 100	94	7	8	41	0.0080
compress 500	107	10	21	38	0.0050
compress 1024	128	12	19	16	0.0040
denoise 0.1	269	2	13	48	0.0095
denoise 0.05	173	3	15	4	0.0025

**Table 3 entropy-23-01481-t003:** Cross-performance PSNR table (exponential signals).

	Train	Block 1/2	Block 1/8	Random 1/2	Random 1/8	Compress 100	Compress 500	Noise 0.1	Noise 0.05
Test									
block 1/2		23.46	22.71	20.21	22.71	23.11	23.23	23.40	21.26
block 1/8		30.73	30.85	27.93	30.85	30.47	30.52	30.51	29.93
random 1/2		34.06	36.11	36.97	36.11	33.44	34.54	33.48	35.00
random 1/8		34.73	37.67	41.61	37.67	33.96	35.62	36.45	35.83
compress 100		17.13	8.56	8.51	8.56	18.98	17.84	15.08	12.78
compress 500		25.87	25.72	17.75	18.56	26.12	25.91	22.11	25.97
denoise 0.1		24.41	21.49	20.02	20.12	25.18	24.51	25.63	24.52
denoise 0.05		29.70	27.61	25.60	26.03	29.83	29.74	28.92	30.14

**Table 4 entropy-23-01481-t004:** Cross-performance PSNR table (RBF signals).

	Train	Block 1/2	Block 1/8	Random 1/2	Random 1/8	Compress 100	Compress 500	Noise 0.1	Noise 0.05
Test									
block 1/2		46.07	43.70	35.94	25.19	39.72	39.17	28.11	45.03
block 1/8		61.33	62.82	62.41	47.80	60.66	59.69	35.28	56.72
random 1/2		67.34	52.59	69.46	65.50	63.34	67.55	29.69	59.88
random 1/8		67.53	53.69	72.25	75.86	63.17	70.12	32.39	58.87
compress 100		39.05	38.87	28.95	18.61	42.94	38.59	30.28	41.86
compress 500		60.52	50.28	54.44	44.42	58.96	61.28	34.94	55.04
denoise 0.1		28.89	34.77	25.73	21.74	32.57	28.31	32.22	34.45
denoise 0.05		36.64	40.26	33.08	29.09	39.26	35.04	27.36	41.23

**Table 5 entropy-23-01481-t005:** PSNR comparisons among baseline models (exponential signals).

	Est.	Opt. DD	Org. DD	Org. DIP	TV Norm	LassoW
Mea.						
block 1/2		22.94	20.99	21.45	22.67	17.72
random 3/4		28.07	27.74	25.85	26.67	17.49
compress 100		18.16	11.44	16.07	16.59	5.69
dno 0.05		29.72	26.81	28.32	29.44	26.86

**Table 6 entropy-23-01481-t006:** PSNR comparisons among baseline models on (RBF signals).

	Est.	Opt. DD	Org. DD	Org. DIP	TV Norm	LassoW
Mea.						
block 1/2		49.06	43.39	29.36	32.58	18.25
random 3/4		65.03	64.69	50.04	42.78	19.47
compress 100		43.32	42.78	17.48	21.21	6.20
dno 0.05		40.77	36.76	29.03	40.95	27.36

**Table 7 entropy-23-01481-t007:** PSNR comparisons among baseline models (NO2 signals).

	Est.	Opt. DD	Org. DD	Org. DIP	TV Norm	LassoW
Mea.						
block 3/4		22.08	16.60	16.94	18.81	16.72
block 1/2		21.51	19.95	19.57	19.04	19.19
block 1/8		27.45	26.25	27.71	24.43	21.59
random 3/4		24.81	24.73	24.81	20.73	17.39
random 1/2		30.00	29.82	27.93	23.01	18.67
random 1/8		40.73	37.90	34.81	27.22	24.47
compress 25		22.61	13.36	10.92	11.99	5.42
compress 256		31.00	30.90	20.97	18.48	16.89
dno 0.1		24.66	20.53	22.64	25.71	21.41
dno 0.05		27.58	26.77	28.24	30.02	22.55

**Table 8 entropy-23-01481-t008:** PSNR comparisons among baseline models (O3 signals).

	Est.	Opt. DD	Org. DD	Org. DIP	TV Norm	LassoW
Mea.						
block 3/4		16.00	15.22	15.95	17.58	15.71
block 1/2		18.76	16.92	18.13	18.64	17.65
block 1/8		25.27	24.91	24.23	21.79	21.53
random 3/4		21.71	21.45	21.10	20.95	16.33
random 1/2		27.66	27.23	24.49	25.19	18.73
random 1/8		38.05	34.43	31.73	33.71	23.65
compress 25		16.70	11.90	8.10	14.72	6.60
compress 256		22.88	22.42	21.80	20.34	15.21
dno 0.1		20.43	22.55	22.45	23.41	20.46
dno 0.05		27.11	28.05	27.63	27.04	20.88

**Table 9 entropy-23-01481-t009:** PSNR comparisons among baseline models (CelebA images).

	Est.	Opt. DD	Org. DD	Org. DIP	TV Norm	LassoW
Mea.						
block 3/4		18.70	16.66	16.23	16.03	12.80
block 1/2		23.65	22.67	20.82	18.03	15.68
block 1/4		27.77	27.43	25.42	22.21	17.35
random 3/4		30.07	28.15	24.59	19.93	16.63
random 1/2		35.65	32.21	29.84	25.98	16.63
random 1/4		39.82	36.63	32.44	31.21	18.09
compress 400		20.11	17.59	18.27	9.35	6.48
compress 4096		27.87	26.06	23.77	14.89	12.57
dno 0.1		38.45	38.22	31.15	24.07	18.47
dno 0.05		49.12	38.01	31.15	28.27	19.77

**Table 10 entropy-23-01481-t010:** Comparisons of various performance metrics and hyperparameter configurations (fastMRI images). The variant of Deep Decoder used here does not have a filtering step, so filter_size is omitted.

Method	VIF	MS-SSIM	SSIM	PSNR	#Channels	#Layers	Input_Size	Filter_Size	Step_Size
Opt. ConvDD	0.9674	0.9429	0.8212	31.8063	252	8	5	3	0.002
Org. ConvDD	0.9599	0.9422	0.8259	31.0642	256	8	4	3	0.008
Opt. DD	0.6021	0.8204	0.6515	28.2947	352	9	18	-	0.002
Org. DD	0.5725	0.8029	0.6529	28.2217	368	10	16	-	0.008
Org. DIP	0.5644	0.8644	0.5163	26.9310	256	16	(640,368)	3	0.008

## Data Availability

This research uses four data sets: (i) Synthetic generated data with code available at https://github.com/ethangela/compressd-sensing (accessed 6 October 2021); (ii) Air quality dataset from http://archive.ics.uci.edu/ml/datasets/Air+Quality (accessed 6 October 2021); (iii) CelebA dataset from https://mmlab.ie.cuhk.edu.hk/projects/CelebA.html (accessed 6 October 2021); (iv) fastMRI dataset from https://fastmri.org/dataset/ (accessed 6 October 2021).

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
