# Peer review of "On Architecture Selection for Linear Inverse Problems with Untrained Neural Networks"

_entropy, 2021, doi:10.3390/e23111481_

Round 1

Reviewer 1 Report

See attached.

Reviewer 2 Report

In the manuscript "On Architecture Selection for Linear Inverse Problems with Untrained Neural Networks", the authors study the effects of a variety of hyper parameters for a deep decoder for solving several linear inverse problems.  The paper provides evidence for the claim that different inverse problem instances should be solved using different choices of hyper parameters.  The authors additionally provide evidence that certain hyper parameters are more robust than others.  The authors also provide a PAC-learning upper bound for risk.  

The manuscript addresses an important and relevant problem: that of attempting to find the most effective untrained neural networks for solving a variety of inverse problems.  

One way to view the paper is its contribution to the subfield of architecture selection.  To make a significant contribution to this subfield, the authors would need to compare their method to other Neural Architecture Search algorithms and demonstrate the superiority of their approach.  Consequently, the contribution to architecture selection is unclear.

Another way to view the paper is through its contribution to the development of priors for linear inverse problems.  The paper's primary contribution on this front is the observation that performance of untrained priors is sensitive to hyper parameter choice (for certain parameters).  This observation confirms what one would presume by default.  The paper can help practitioners by informing them of which hyper parameters are more important than others, and the algorithms to tune hyper parameters could be useful to practitioners.  That said, the specific algorithms tuned in the paper are presented for limited synthetic problems, and the paper would be significantly strengthened if they used their techniques to get a method for a more realistic problem and compared to state of the art methods for those problems.   

Summary: To have a significant contribution for architecture selection, the paper needs to compare to other methods.  To have a significant contribution for specific image priors, the paper should evaluate the performance of their methods on significant applied problems (such as superresolution, MRI, etc) and compare to other approaches.  Without those two, the primary contribution of the paper is an algorithm to help researchers working with untrained networks optimize over hyper parameters, which is somewhat incremental. 

Reviewer 3 Report

The article is not suitable for publication due to following reasons:

  1. Introduction is below average. Although the it describes the background by illustrating the general area of research and also mentions the importance of selected research area by highlighting its critical factors, however, it does not present the research problem by explaining what is really required in the selected research area. Similarly, it briefly overviews the existing practices.  However, the limitations of existing practices  are not identified and hence does not highlight the research gape. It implies that how the proposal will solve the limitations is not clear. Moreover, the novelty of proposed solution is not highlighted. Other details about the proposal such as validation (description of case studies and/or benchmarks) are also missing. Finally the outcomes and impact of solution are not clearly articulated.
  2. The proposed method is not presented in an appropriate way. Some important details are missing. The design must include the structure and behaviour details.
  3. The following details about the validation are not provided: Experimental setup, description of tools and particular settings  used in the experiment, description of benchmarks and case study, experimentation procedure.
  4. The final major limitation is the comparison of achieved results with state-of-art solutions, using tables, graphs and any other methods. Good data visualization techniques have not been used. It is also important to use various performance parameters for comparisons. Each performance parameter must be defined explicitly before its use. You should also provide the justification and motivation behind the selection of each performance parameter. 

Round 2

Reviewer 1 Report

The paper can be accepted for publication in the present form.

Author Response

Thank you again for your feedback.

Reviewer 2 Report

I appreciate the revisions that the authors have put in place in response to my feedback from before.  I do believe that some researchers will find the conclusions of this paper useful and may build better models for solving inverse problems as a result.  I think the paper can be published in its current form.

Author Response

Thank you again for your feedback.

Reviewer 3 Report

Section 3 (experiments) is required to be significantly revised. This section is further divided into 7 subsections. An introductory paragraph in the beginning of section 7 is written, however, it fails to really provide a clear connection between theses 7 subsections.

The obtained results have not been compared with state-of-the art.

The outcomes from experiments have not been discussed in terms of their significance.

A lot of tables and figures have been presented, however, the contribution is not evident from results.

The introduction section of the revised article has now provided the required information. However, the presentation and organization is really unconventional and difficult to comprehend. Generally, the introduction is described in the form of following paragraphs:

Paragraph 1: It describes the background by illustrating the general area of research. It also mentions the importance of selected research area by highlighting its critical factors.  

Paragraph 2: It presents the research problem by explaining what is really required in the selected research area. Here, the authors can clearly state that why Untrained Neural Networks are needed for Linear Inverse Problems, particularly for Architecture Selection.

Paragraph 3: It briefly overviews the existing practices (without details).  The details should be described in a separate section. Limitations of existing practices are identified and hence highlighting the research gape.

Paragraph 4: What you have proposed (briefly describe). How it will solve the limitations highlighted in paragraph 3.  The novelty of proposed solution should be highlighted.

Paragraph 5:  How u have validated your solution (description of case studies and/or benchmarks. Why these benchmarks/case studies are important and interesting. What is the motivation behind the selection of these benchmarks) 

Paragraph 6: Significance and impact of your solution.

Paragraph 7: Organization of ur article. 

In addition to the introduction section, there should be a dedicated section which can provide the appropriate background knowledge and a detailed review of existing works. This section can be divided into 2 sub-sections:

--An introductory paragraph can be written to justify that why the background information in section 2.1 is really needed to understand this article. 

---Section 2.1 can be divided into further subsections (if needed). 

---Then Section 2.2 should describe the existing methods (at least 7 to 8) from reputed journals. You can classify the existing methods in some categories. Here you can also define various attributes and compare the existing methods in terms of those attributes. 

---Finally Section 2 must be closed with a concluding paragraph , summarizing the limitations in existing methods highlighted in Section 2.2. Here, you should also state that how your solution will address the shortcomings

Author Response

We appreciate these detailed thoughts on the structure and content of our paper.  It is evident that we have some differences in style, preference, and expectations, possibly stemming from coming from different research backgrounds.  Despite this, we believe that the current format of our paper is suitable and matches similar kinds of papers in the existing literature, many of which we cited.  We have done some final editing of our paper in this revision, but kept the structure and content largely similar.  We leave the decision of whether this format/content is sufficient up to the editor, and we will make further (possibly non-minor) edits at their request.  Thank you again for your comments.